# Dexibuprofen Therapeutic Advances: Prodrugs and Nanotechnological Formulations

**DOI:** 10.3390/pharmaceutics13030414

**Published:** 2021-03-19

**Authors:** Anna Gliszczyńska, Elena Sánchez-López

**Affiliations:** 1Department of Chemistry, Wrocław University of Environmental and Life Sciences, Norwida 25, 50-375 Wrocław, Poland; 2Department of Pharmacy, Pharmaceutical Technology and Physical Chemistry, University of Barcelona, 08028 Barcelona, Spain; 3Institute of Nanoscience and Nanotechnology (IN2UB), University of Barcelona, 08028 Barcelona, Spain

**Keywords:** dexibuprofen, NSAIDs, enantiomer, drug delivery, prodrugs

## Abstract

*S*-(+) enantiomer of ibuprofen (IBU) dexibuprofen (DXI) is known to be more potent than its *R*-(−) form and exhibits many advantages over the racemic mixture of IBU such as lower toxicity, greater clinical efficacy, and lesser variability in therapeutic effects. Moreover, DXI potential has been recently advocated to reduce cancer development and prevent the development of neurodegenerative diseases in addition to its anti-inflammatory properties. During the last decade, many attempts have been made to design novel formulations of DXI aimed at increasing its therapeutic benefits and minimizing the adverse effects. Therefore, this article summarizes pharmacological information about DXI, its pharmacokinetics, safety, and therapeutic outcomes. Moreover, modified DXI drug delivery approaches are extensively discussed. Recent studies of DXI prodrugs and novel DXI nanoformulations are analyzed as well as reviewing their efficacy for ocular, skin, and oral applications.

## 1. Introduction

Nonsteroidal anti-inflammatory drugs (NSAIDs) are members of a drug class that reduces pain, decreases fever, prevents blood clots, and decreases inflammation [1,2]. In general, NSAIDs are characterized by a high degree of protein binding and small volumes of distribution [3]. However, in this family, differences in clearance and variability in half-life are of special relevance. Among of all NSAIDs, ibuprofen (IBU) is one of the most widely and frequently employed therapeutic agents with the lowest toxicity. IBU was initially introduced in the UK in 1960s and afterwards during the 1970s worldwide as a prescription-only medication with recommendation dose 2400 mg/day (or higher in the USA) [4]. Nowadays, IBU is widely used in many countries and presents minimum side effects involving gastric damage observed in many other NSAIDs. Over the last three decades, the over-the-counter (OTC) IBU-containing preparations and the generic preparations sold by prescription and non-prescription are used as a NSAID with analgesic and antipyretic properties [5]. They are used for the treatment of osteoarthritis, rheumatoid arthritis and in a wide variety of other painful conditions [6]. These therapeutic effects come from activity of IBU to block prostaglandin synthesis by a non-selective, reversible inhibition of the cyclooxygenase enzymes COX-1 and COX-2 [7]. IBU achieves high plasma protein binding and low distribution volume, but it possesses the ability to be accumulated in effective amounts in compartments where inflammation occurs. These compartments, such as cerebrospinal fluid (CSF) where there is need for anti-inflammatory/analgesic activity, are the target of IBU.

In most of the commercially available preparations, IBU occurs only as a diastereoisomeric mixture which comprises equal quantities of *R*-(−)-ibuprofen and *S*-(+)-ibuprofen. Dextrarotatory isomer of ibuprofen dexibuprofen (DXI) is the pharmacologically effective enantiomer, which was for the first time launched in Austria in 1994 [8]. Racemic ibuprofen and DXI differ in their physical, chemical, and pharmacological properties as well as their metabolic profile [5,9]. In the last five years, 4836 patients have been exposed to DXI in clinical trials and post marketing surveillance (PMS) trials. Only in 3.7% of patients have adverse drug reactions been reported and three serious adverse drug reactions (0.06%) occurred [5]. It has been proven in the in vitro model that dextrarotatory isomer exhibits about 160-times higher activity in prostaglandins inhibition in comparison to enantiomer (*R*). Other studies of thromboxane generation in clotting blood also confirmed higher activity of enantiomer *S* than racemate [10]. Therefore, it would be highly advantageous to use DXI as a pain reliver. Especially, due to the fact that while DXI effectively inhibits the activity of COX-1 and COX-2, the enantiomer (*R*) demonstrates the inhibition only towards COX-1 and it is worth noting that it is responsible increasing the side effects in the gastrointestinal tract [11].

*rac-*Ibuprofen undergoes an unusual metabolic fate because inactive *R*-(−)-ibuprofen is enzymatically converted to the therapeutically active enantiomer *S*-(+) [12]. This may contribute to variability in analgesia including delayed onset of activity and may explain the poor relationship observed between plasma concentrations of IBU and clinical response for acute pain and rheumatoid arthritis [13]. According to published data in humans, average 50% of enantiomer (−) is metabolically converted via catalytic activity of fatty acyl coenzyme thioesterase to more active enantiomer (+) [14,15] in the intestinal tract and liver after oral absorption [16,17]. However, it should be pointed out that the level of metabolic inversion of IBU may vary between 35 and 85% depending on the formulation type, condition of the liver, the intake of medicines, and the state of the disease [18,19]. During this process, the first step is the activation of *R*-(−)-ibuprofen in the presence of coenzyme A (CoA), adenosine triphosphate (ATP), and Mg^2+^ (Figure 1).

Next, the AMP-derivative is esterified with coenzyme A (CoA) by the action of acyl-CoA synthetase. *R*-(−)-ibuprofen-CoA undergoes epimerization via the actions of the -methylacyl-coenzyme A racemase (encoded by gene AMACR) [20] to form *S*-(+)-ibuprofen-CoA, which is then hydrolyzed by a hydrolase to form *S*-(+)-ibuprofen [5]. The key step of this process is the removal of the substrate -proton followed by non-stereoselective reprotonation. The process of inversion can occur pre-systematically in the gut [21] as well as in the liver [22].

Thioesters formed from coenzyme A and ibuprofen are subsequently incorporated into triglycerides (TAG) or phospholipids (PLs), forming hybrids that may have an influence on the function of the cell membranes. However, it is described in the literature that only *R* enantiomer of IBU has been detected in the human adipose tissue and its elimination half-life (t_1/2_) is about seven days. It was proven that oral administration of pure enantiomer *S* allows a stronger analgesic effect in a shorter time compared with racemic mixture. Higher effectiveness therapy of *S*-(+)-ibuprofen and significantly reduced side effects of its administration have been demonstrated on the group of 1400 patients [21,23].

As a general approach, it was firstly claimed that a dose of 1:0.75 (ibuprofen and DXI, respectively) would be necessary in order to obtain similar pharmaceutical efficacy with DXI than IBU [5]. Regarding this aspect, several clinical trials have been performed using a dose ratio of 0.5:1 assuming the fact that 50% of the dose contained in IBU corresponds to DXI. In these clinical trials, DXI therapeutic efficacy was similar to IBU. Other studies carried out comparing DXI with other NSAID such as Diclofenac pointed out that regarding the maximum daily dose (MDD) using 75% of the MDD of DXI, the same efficacy as 100% of Diclofenac was achieved [8]. Moreover, regarding DXI interactions with other compounds, the data obtained to date indicate that no influence was found on bioavailability with or without food intake. So far, DXI has demonstrated clinical efficacy in rheumatoid arthritis, ankylosing spondylitis, osteoarthritis of the hip, osteoarthritis of the knee, lumbar vertebral syndrome, distortion of the ankle joint and dysmenorrhoea. The tolerability was better than other NSAIDS since racemic ibuprofen showed a 30% increasing incidence of adverse drug reactions and diclofenac a 90% higher than DXI. Therefore, DXI would potentially combine the high efficacy of diclofenac with the good tolerability of ibuprofen [8]. In addition, DXI has also been compared with Celecoxib, a selective inhibitor of cyclooxygenase-2 (COX-2) for the treatment of hip osteoarthritis. DXI demonstrated equal efficacy to Celecoxib and a comparable tolerability profile [24].

DXI has a slower dissolution rate in the simulated gastric and enteric juices compared with the racemic ibuprofen and displays improved oral bioavailability. Pure DXI exhibits more advantages than its former, but due to its weak acidity (pK_a_ = 5.2) unfortunately its bioavailability is relatively low due to the limited solubility in acidic media of the stomach. According to Biopharmaceutics Classification System (BCS), DXI is classified as the poorly water-soluble drug but well-penetrating substances substance (group II). Therefore, its bioavailability is limited by its poor dissolution, so its absorption rate is similar to that of dissolution. These physicochemical properties of DXI result in many limitations in in vivo models like incomplete release, food interactions, or high inter-subject variability. To overcome these mentioned limitations, many attempts have been made so far. As the method for improvement, the synthesis of effective prodrugs of DXI or preparation dedicated to different application routes such as ocular, dermal or oral routes formulations have been proposed and this will be the subject of this review. These applications will be discussed throughout the review and novel drug delivery forms containing DXI will also be analyzed.

## 2. Dexibuprofen Main Pharmacological Properties

Dexibuprofen, *S*-(+)-ibuprofen, (2*S*)-2-(4-isobutylphenyl)propionic acid (DXI), is the dextrarotatory, more pharmacologically effective and more potent enantiomer of ibuprofen [25]. Enantiomer *S*-(+) IBU exerts its effect by suppressing the prostanoid synthesis in the inflammatory cells mainly via inhibition of the COX-2 isoform, causing analgesic, antipyretic, and anti-inflammatory effects [10,26]. *S*-(+)-ibuprofen is less toxic than the enantiomer *R*. DXI possesses higher water solubility (68.4 mg/L) than ibuprofen (21 mg/L) and a lower melting point (49–53 °C vs 75–77.5 °C, DXI and ibuprofen, respectively). Moreover, DXI also presents increased stability, a slower dissolution rate, and improved bioavailability [26].

DXI is currently available in the European Union (EU) currently manufactured as tablets against pain and inflammation containing 400 mg of DXI [27]. Moreover, several clinical trials are underway. In the US data base, two Phase I clinical trials with a currently unknown status have been approved to evaluate the safety of DXI 300 mg and 200 mg after a single and multiple dose oral administration in healthy participants (NCT0295651, NCT02956525) [28,29]. Moreover, a completed phase I clinical trial has also been performed by Gebro Pharma GmbH to assess DXI effects on aspirin-treated volunteers (NCT00442585) [29]. Furthermore, in 2010 a phase 3 clinical trial was completed aimed at studying the efficacy and safety of DXI syrup with fever due to common cold (NCT00812422) [30]. A phase IV postmarked clinical study completed in 2012, assessed and compared the tolerability profile of DXI Gebro 400 mg powder for oral suspension against ibuprofen 400 mg in patients with painful osteoarthritis of the hip or knee (NCT01066676) [31]. In the EU database, a completed prospective phase 3 clinical trial is registered, aimed at investigating the safety, tolerability and efficacy of DXI Gebro 400 mg powder for oral suspension (test) compared to ibuprofen 400 mg powder for oral suspension (reference) in patients suffering from osteoarthritis of the hip or knee, showing positive results for DXI against its racemic counterpart (2009-010719-33) [32].

The majority of DXI formulations are intended to be administered orally to reach systemic circulation [33]. After oral administration, DXI is rapidly and extensively absorbed from the upper GI tract with its peak values in the plasma and serum at approximately t_max_, ~ 1–2 h depending on the specific oral formulations [5]. Moreover, it has been reported that unbound DXI concentrations show linear pharmacokinetics at commonly used doses. Absorption of DXI occurs on a principle of a passive process where its crystal form may play a crucial role for the interpretation of experimental and clinical data. It should be noted that administration of DXI with meals delays the time to reach maximum concentration (from 2.1 h after fasting conditions to 2.8 h after non-fasting conditions) and decreases marginally the maximum plasma concentration (from 20.6 to 18.1 g/mL). However, food has no effect on the extent of absorption. A linear dose-response relationship was shown over the dose range from 200 to 400 mg [14,21,34]. 

Phase I metabolism of both enantiomers of IBU involves hydroxylation of the isobutyl chains to 2-hydroxy- and 3-hydroxyibuprofen and subsequent oxidation of the latter to 3-carboxyibuprofen and *p*-carboxy-2-propionate (Figure 2). Moreover, some reports confirm DXI preference against the R-enantiomer in both phase I and II metabolism [35].

In addition, protein binding has been determined to be enantioselective. In the case of IBU, the *R*-enantiomer has shown that the binding constant between proteins and the *R*-enantiomer high-affinity protein binding site is 2.6-fold higher than for the *S*-enantiomer (DXI). This decreased binding of DXI may cause higher transfer into blister fluid or synovial fluid, where it elicits the pharmacological activity [35,36].

The major metabolites are 2-hydroxy-ibuprofen and carboxy-ibuprofen (and their corresponding acyl glucuronides) which account in the urinary excretion for around 25% and 37% of an administrated dose, respectively [14,21,34], Metabolism of *S*-(+)-ibuprofen is predominantly catalyzed via cytochrome isoform CYP-2C9, whereas its *R*-(−) antipode is formed more via isoform CYP-2C8, [36,37]. No difference in the pharmacokinetics of *R*-(−) and *S*-(+) enantiomer of IBU have been found related with gender with the exception of the volume of distribution V_D_/F being about twice that in adult females compared with males [38].

## 3. Dexibuprofen Prodrugs

NSAIDs are one of the therapeutic groups most widely used, prescribed, and over the counter medications, which administration leads to serious gastrointestinal (GI) complications. These side effects of NSAIDs associated with long term of their oral use, are attributed to the presence of free carboxylic group (-COOH) in their structures. Therefore, many attempts have been made, especially in the last decade, to mask this free carboxylic group and synthesized prodrugs of NSAIDs [39,40] and also overcome pharmaceutical, pharmacokinetic, and pharmacodynamic barriers. In the case of DXI, a number of functional groups have been employed for the preparation of its prodrugs that are trying to reduce its tendency to create peptic ulceration and GI track bleeding. Its conjugates with polymers, amino acids, and ethanolamine have been obtained and proposed as the forms that can constitute an effective approach to solve the problems, slow down DXI action achieving a prolonged drug effect, avoiding adverse effects and increasing therapeutic adherence.

Ashraf et al. synthesized two series of prodrugs of DXI, amides and esters (Figure 3). To synthesize amide prodrugs **1a–e** five amino acids were selected: glycine, alanine, valine, leucine, and phenyl alanine. The ester prodrugs **2a–e** were obtained in the reaction of DXI with different alcohols: isopropyl, *n*-butyl, isobutyl, benzyl, and glycerol. The solubility studies showed that synthesized series of DXI derivatives have more lipophilic character and exhibit low protein binding that lead to an increase in their availability in plasma for hydrolysis and reduce the required active dose. All studied prodrugs showed hydrolysis rate in simulated intestinal fluid (SIF pH 7.4) in 80% plasma. It was also proved that DXI in the form of prodrugs prevents the accumulation of the drug in gastric mucosa, decreasing the possible gastrointestinal irritant without loss of pharmacological effect [41].

These findings were deeply extended and DXI amide analogues with alkyl/aryl substitution were evaluated as anticancer agents towards breast carcinoma cell line (MCF-7). It was determined that the presence of chlorine atom in the phenyl ring has the strongest effect on biological activity and the amides bearing 2,5-dichloro and 2-chloro substituted phenyl ring turned out to be the most active and exhibited 100% inhibition of the tumor growth. Dichloride substituted amide *N*-(2,5-dichlorophenyl)-2-(4-isobutylphenyl)-propionamide was active even at lower concentration doses IC_50_ = 0.01 ± 0.002 μM than doxorubicin IC_50_ = 0.04 ± 0.006 μM [42]. Amide derivatives of DXI **1b,d,e** were also studied in the aspect of how they interact with DNA using spectroscopic, electrochemical, and molecular docking techniques under simulating physiological conditions at stomach and blood pH, 4.7 and 7.4, respectively at temperature 37 °C. The authors observed spontaneous interaction of all the compounds with DNA via intercalation and external bindings [43].

High solubility in organic solvents and high hydrolysis rate in 80% human plasma have been reported also for amide prodrugs of DXI **3–6** synthesized in the reaction of DXI acid chloride with methyl esters of L-tryptophan, L-phenylalanine, glycine and l-tyrosine (Figure 4) [44]. Prodrugs **3–6** were studied on their various physicochemical as well as pharmacological properties on the group of Wistar albino rats. Amides were found to be significantly less ulcerogenic with much higher anti-inflammatory activity (64.5–77.3%) than that reported for the parent drug (43.3%). The maximum of their activity was observed at 6 h and was practically constant up to 8 h.

Rasheed and colleagues synthesized DXI-dextran prodrug **7** for oral administration aimed at improving DXI aqueous solubility, increasing its therapeutic efficacy, and reducing gastrointestinal side effects (Figure 5) [45]. They used DXI acyl imidazole derivatives and dextran of different molecular weights (10,000–20,000 Da) to obtain DXI-dextran prodrugs with the substitution degree between 15.6–16.5%. Obtained prodrugs were characterized by faster hydrolysis at pH 9.0 than pH 7.4. In the in vivo study carried out on albino rats, DXI-dextran did not show improvement of the analgesic activity against DXI but showed a slight increase in the anti-inflammatory potential. The results also revealed that ulcerogenicity of DXI-dextran prodrugs in stomach was reduced compared with free form of this drug and clearly indicate that side effects of DXI were reduced after conjugation with dextran [45]. Recently, Preethi also esterified DXI with dextran (M_w_ 60,000–90,000 Da) in the one pot chemical reaction using *N*,*N*-carbonyldiimidazole [46]. This prodrug shows a log P of 5.4 and in artificial intestinal fluid was significantly hydrolyzed (99.53%) by following first-order kinetics with 85.9 min half-life. During the preclinical experiments in the in vivo models on the Wistar rats, DXI-dextran conjugate showed superior analgesic, anti-inflammatory, antipyretic activities in [46].

Other groups have also synthetized DXI prodrugs forming DXI-antioxidant conjugates with menthol, sesamol, and umbelliferone **8–10** aimed at reducing gastrointestinal side effects (Figure 6). In order to achieve this purpose, ester analogues of DXI were obtained [47]. The prodrugs were stable in the stomach whereas in plasma they underwent hydrolysis processes releasing DXI. Since minimum hydrolysis was observed at acidic pH (1.2), this may indicate that DXI-ester prodrugs are less irritating that DXI. Moreover, these results were confirmed in vivo performed on male and female mice orally administrated with the synthesized prodrugs [47]. Preclinical experiments showed that these DXI-ester prodrugs caused a significant increase in anti-inflammatory, analgesic, and antipyretic activity [47].

NSAIDs have been reported in the literature as active molecules in the treatment of breast cancer cell line MCF-7 and colon cancer cells HT-29 [48,49,50,51]. Based on these findings, DXI was covalently linked via alkylene spacer units to a riboflavin derivative **11a–e** (Figure 7) due to well-known ability of this moiety to reduction of cancer chemotherapy toxicity and protection against oxidant-mediated inflammatory organ injury [52]. Synthesized conjugates of DXI with 2′,3′,4′,5′-tetraacetylriboflavin (TAR) exhibited comparable antiproliferative activity when used in the treatment of chemotherapy 5-fluorouracil (5-FU) towards both tested cancer cell lines MCF-7 and HT-29 with IC_50_ values in the range of 7.8–14.9 M. The authors have interpreted observed cytotoxicity of synthesized prodrugs by intracellular release of DXI or by the fact that conjugates are able to act not only as drug delivery agents but independent active molecules [53]. 

Clinical application of DXI and other NSAIDs in the treatment of neurodegenerative disorders is strongly limited by the lack of their brain accessibility. In order to overcome limited distribution of DXI in the central nervous system (CNS), Zhang et al. have designed and synthesized five prodrugs **12a–e** (Figure 8) [54]. These compounds were obtained via esterification reaction of the carboxy group of DXI with ethanolamine with substituted amino group by methyl, ethyl and acetyl groups and their brain-targeting efficiency was evaluated on male Sprague-Dawley rats. In the biodistribution study, the brain-to-plasma concentration ratios were 9.31- to 17.0-fold higher for DXI derivatives modified by ethanolamine related structures than for DXI after 10 min from their intravenous administration. The mechanism responsible for this enhancement was next studied by Li et al. using an in vitro blood brain barrier (BBB) model and in situ perfusion technique [55]. DXI prodrugs with a primary and secondary amine enter via passive diffusion, whereas prodrugs with tertiary amine modifications seem to enter using an active process that is energy and pH dependent but was independent of sodium or membrane potential [55]. Since the experiments were carried out in vitro, DXI prodrug with a tertiary amine could constitute a potential suitable approximation in order to increase transport across the BBB using the pyrilamine-sensitive H^+^/OC antiporter. However, in vivo results will be necessary to confirm its efficacy [55].

New approach for the synthesis of prodrugs of DXI was presented by Kłobucki and coworkers [56]. They obtained a series of novel phosphatidylcholines **13**–**16** containing dexibuprofen in their structures (Figure 9), which were subsequently evaluated towards two cancer cells (HL-60 and Caco-2) and normal iporcine epithelial intestinal (PEC-J2) cells. Lysophosphatidylcholine containing DXI in the *sn*-1 position **16** and asymmetrically substituted phosphatidylcholine **14–15** were characterized by lower toxicity than DXI against HL-60 cells. The obtained results confirmed presented in the literature data indicate that phospholipid derivatives of active compounds are less cytotoxic than the compounds themselves and that phospholipids could be used also as an effective carriers of NSAIDs [57,58].

However, despite DXI prodrugs attempts of some authors, a high number of modifications have not been attempted with this compound. This opens a window for future research of more effective and less damaging DXI prodrugs able to undergo clinical practice.

## 4. Novel Strategies for Dexibuprofen Drug Delivery

### 4.1. Novel Strategies for Ocular Dexibuprofen Delivery

Ocular drug delivery is one of the most interesting and challenging endeavors facing pharmaceutical scientists [59]. NSAIDs have been used and widely investigated for ocular drug delivery. This is a group of important modulators of ocular inflammatory reactions which act by inhibiting the biosynthesis of prostaglandins and inhibiting intraoperative miosis during cataract surgery, reducing the vascular permeability of the blood-ocular barrier, and modifying inflammation. However, they are not capable of inhibiting the actions of protaglandins once formed [59].

Different approximations have been carried out in order to use DXI for ocular drug delivery. Mixing of DXI with surfactants or other compounds to increase DXI solubility is a practical approach, such as the research carried out by Zhong-Kun and colleagues who developed arginine-DXI for ocular administration as eye drops. However, clinical results obtained have not been reported [60].

More sophisticated methods to increase DXI drug delivery using topical administration are the development of polymeric nanoparticles. Polymeric colloidal particles range in sizes from 10 to 1000 nm and, among all, the most widely used polymer its PLGA [61]. In this sense, Sánchez-López and colleagues developed a formulation loading DXI into PLGA PEGylated NPs to be administered as eyedrops [62]. DXI demonstrated to be non-irritant neither in vitro nor in vivo. Moreover, the authors achieved a sustained DXI release as well as increased therapeutic efficacy against prevention of corneal inflammatory processes. Moreover, they also assessed several PLGA NPs with different PEGylation degrees and several surfactants adapted for ocular drug delivery using DXI as a model drug. In this sense, DXI potential to be used for drug delivery was demonstrated and its ability to decrease ocular inflammation more effectively using PEGylated PLGA nanoparticles was confirmed [63]. Other authors have also used PLGA nanoparticles for ocular drug delivery of NSAIDs, thus confirming the therapeutic potential of these biodegradable nanoparticles [64,65]. Moreover, lipid carriers also constitute a suitable approach for NSAIDs ocular administration [66]. In this sense, several authors have encapsulated ibuprofen into lipid nanocarriers [67]. However, to our knowledge no lipid nanoparticles containing DXI have been reported.

### 4.2. Novel Strategies for Skin Dexibuprofen Delivery

Skin provides protection from environmental injuries, prevents microbial invasion, regulates temperature, and maintains hydration [68]. Inflammatory skin associated disorders are one of the most common pathologies that affect this tissue [68]. Therefore, NSAIDs are widely used for topical administration. In this sense, recent drug delivery formulations encapsulating DXI in order to increase its solubility and skin permeation have been developed.

One of the most common forms of transdermal drug delivery are transdermal patches (Table 1). With this purpose, DXI transdermal patches have been formulated as anti-rheumatic medication [69]. The patch matrix was made of ethyl cellulose and polyvinylpyrrolidone, a plasticizer (di-*N*-butyl phthalate), and a permeation enhancer (almond oil). The patches demonstrated a uniform thickness and a high DXI uptake [69]. The formulation patch was compared with commercial DXI (previously dissolved tablets) in vivo, obtaining a plasmatic concentration 1.9 times higher with DXI patches than with the oral tablets [69].

Other authors have also developed DXI transdermal patches for controlled DXI delivery [70]. In this sense, Ramzan Ali and colleagues developed a reservoir-type transdermal patch for a controlled delivery of DXI and evaluated its in vivo anti-inflammatory activity performing the carrageenan hind paw edema test in Albino Wistar rats. The formulations were composed of ethyl oleate, Tween 80, and PG and the revoir patch compartment was filled with them. The formulations did not cause any irritation signs and were able to deliver DXI across the skin, being effective against inflammation [71].

Another strategy to increase adhesivity to skin pharmaceutical formulations is the development of drug-dispersed gels [71]. Using this strategy, Thiruppathi and colleagues prepared a DXI emulsion and incorporated it into an aloe vera gel base, obtaining high loading efficiency (78%) [71]. Therefore, the authors combined an emulsion and a gel creating an emulgel and compared its effects with traditionally used diclofenac gel. In vivo anti-inflammatory studies were performed, showing the DXI emulgel’s superior edema inhibition and higher analgesic properties. Emulgels have emerged as a promising drug delivery system for the delivery of hydrophobic drugs. Other authors prepared an emulgel based on Carbapol 940. There were two natural penetration enhancers used, clove oil and mentha oil. In accordance with the results obtained by other authors using DXI emulgels, the formulation shows superior anti-inflammatory and analgesics effects than topical marketed diclofenac [72].

Moreover, a patent developing DXI hydrogels containing a pH modifying agent, antioxidant, and water miscible solvent has been published using HPMC as a gelling polymer. These gels were stable for 3 months under accelerated stability conditions (40 °C/75% residual humidity) [73].

### 4.3. Novel Strategies for Oral Dexibuprofen Delivery

NSAIDs have been clinically used for systemic drug delivery after oral administration for several purposes including pain relief, analgesia, and antipyretic effects. Moreover, they have also been used for local delivery of drugs through the oral mucosal tissue.

In order to achieve DXI systemic delivery, several strategies have been employed (Table 2). Using a practical approach, DXI chewable tablets in order to enhance DXI dissolution rate were developed by Neem and colleagues [74]. In a more sophisticated manner, Li and colleagues developed acidic montmorillonite composites able to encapsulate DXI (298 mg/g) [75]. Montmorillonite is a clay mineral, used in industrial and pharmaceutical fields due to its swelling and adsorption properties. DXI composites show an average size on the micrometric range (0.96 ± 0.12 μm) and a negative surface charge (−26.5 ± 0.94 mV). DXI composites from acidic montmorillonite composites were released within 12 h (against DXI which was released in 4 h) in simulated intestinal fluid. In addition, DXI composites also show to retard drug release in vivo in a rat preclinical model and, at the same time, increase DXI bioavailability compared to commercial DXI suspension [73].

In order to enhance aqueous solubility of DXI, hydroxypropyl-β-cyclodextrin (HPβCD) hydrogel nanoparticles were designed by Khalid and colleagues. Hydrogels consist of the formation of a three-dimensional matrix of physical or chemical cross-linked polymers that take up water. The HPβCD hydrogel nanoparticles encapsulating DXI were able to increase DXI aqueous solubility forming a stable polymeric network. The average size of nanoparticles was 287 nm able to release DXI preferentially at pH 1.2 and 6.8. Moreover, no acute toxicity signs were reported after in vivo preclinical administration [75]. In a similar study of the same research group, Khalid and colleagues developed hybrid nanogels. Nanogels are a type of hydrogels formed by nanoscopic networks. Therefore, nanogels combine hydrogels and nanocarriers advantages. In the study, HPβCD hybrid nanogels in order to enhance the solubility of DXI were carried out. The HPβCD nanogels were able to encapsulate DXI, showing a significant DXI release in aqueous media. In vivo preclinical studies show no toxic effects of DXI, confirming the biocompatibility of the developed nanogels. However, in vivo therapeutic effects have not yet been reported [76].

Other authors have also improved DXI solubility by developing a DXI-loaded solid dispersion using novel techniques such as spray-drying. The authors formulate DXI for immediate release by combining DXI with Poloxamer 407, HPMC, and sodium lauryl sulphate. The formulation shows higher plasma concentration than DXI powder. Furthermore, DXI formulation for prolonged release was developed using DXI, ethylcellulose, HPMC, and magnesium stearate and encapsulating the powder obtained by spray dry into gelatin capsules. After in vivo oral administration in an animal model, this formulation enhanced DXI residence time and also its maximum plasma concentration against DXI powder [77]. Other authors have also recently developed DXI sustained release supermicro-pellets-based dry suspensions using a bottom spray fluid bed with suitable properties for prolonged drug release [78].

Another strategy is the combination of hydrogels, cyclodextrins, and gelatin capsules in order to obtain DXI mucoadhesive oral administration for systemic delivery. The authors were able to produce a formulation with a pH dependent swelling capacity being its maximum swelling at pH 1.2. Moreover, the formulation also shows a controlled drug release in a pH responsive manner. These results are promising for further in vivo studies due to the combination of several strategies leading to a potentially useful DXI carrier [77]

In order to encapsulate DXI, different types of polymeric nanoparticles have also been developed. PLGA Pegylated nanoparticles for DXI brain delivery were developed for oral administration with a particles size below 200 nm and a low polydispersity index (below 0.1). Slower DXI release from PLGA PEGylated nanoparticles was achieved in vitro compared with DXI. Decrease gastric damage of DXI-loaded nanoparticles was also demonstrated in a mice model. Using a transgenic mice mode for Alzheimer′s disease, the developed drug delivery systems show capacity to reduce brain inflammation and also the number of β-amyloid plaques more effectively than DXI. Additionally, DXI nanoparticles reduced memory impairment associated with transgenic mice [79].

Other polymeric nanoparticles have been also developed for DXI drug delivery such as chitosan-based nanoparticles. Chitosan is a natural polymer, widely used for drug delivery due to its biodegradability and also to its positive surface charge. DXI-chitosan developed formulations show an average particle size of 437.6 nm and high drug loading. Moreover, drug release was prolonged up to 24 h. Although these nanosystems seem promising for DXI drug delivery, to our knowledge no in vivo data have been provided up to date [81].

Another widely used polymer for systemic administration for oral drug delivery is Eudragit. Microparticles containing DXI were developed showing enhanced rheological properties with an entrapment efficacy of 70%. (103) In vitro DXI studies verified the gastro-resistant ability of micro particles [81].

Another example using Eudragit, Bertalero and colleagues developed DXI-Eudragit-TPGS solid dispersion nanoparticles (<300 nm) by using the supercritical antisolvent technique. This drug delivery system improved in vitro DXI dissolution and oral absorption. Moreover, plasma concentration and DXI bioavailability were also increased [82].

In addition, DXI polymeric micelles prepared using Pluronic have also been studied for oral availability enhancement of DXI. Clinical pharmacokinetic studies were performed in order to demonstrate enhanced oral bioavailability of the marketed formula, enhancing the delivery of DXI in human patients [83].

## 5. Conclusions

DXI is the active enantiomer of the widely used IBU. Due to enantioselective characteristics, DXI possesses some unique properties that are able to modify its pharmacokinetic profile against the racemic mixture.

It is currently on the market in the form of tablets and several clinical trials are underway with this compound. Moreover, several investigations are aimed at enhancing DXI water solubility in order to achieve higher plasmatic concentrations and decrease gastric side effects after oral administration. In this sense, DXI prodrugs have been recently synthetized showing successful preclinical results. The majority of these prodrugs were aimed at masking the DXI free carboxylic group by adding different chemical groups by showing less gastrointestinal adverse effects.

Furthermore, drug delivery formulations for oral and skin transdermal permeation have also shown successful results. These formulations include different types of polymers and other substances intended to deliver DXI slowly to the target site. In this area, the use of nanotechnology becomes of special relevance and several attempts have been undertaken to improve DXI release profile using nanoscopic systems.

As a conclusion, DXI is being currently explored for several pharmaceutical applications due to its great potential over its widely known racemic mixture. Moreover, improvement aimed at increasing DXI elimination providing a sustained drug release have been carried out either by synthetizing DXI prodrugs or by using nanotechnological approaches to enhance drug delivery.

## Figures and Tables

**Figure 1 pharmaceutics-13-00414-f001:**
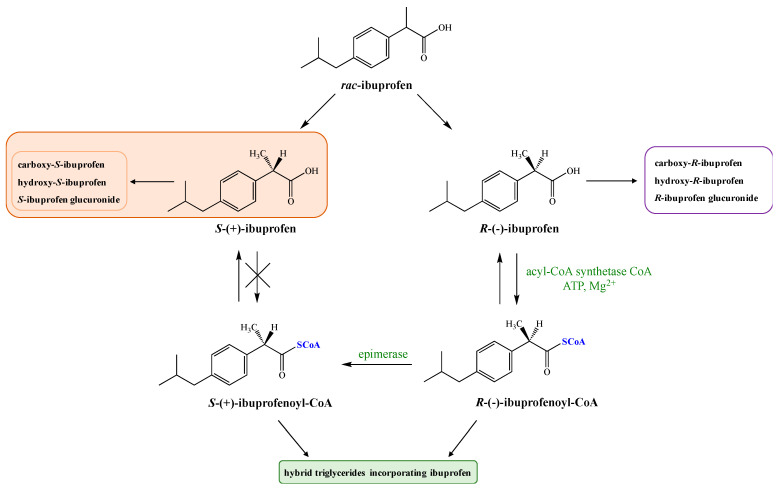
Metabolic conversion of *R*-(−)-ibuprofen to *S*-(+)-ibuprofen.

**Figure 2 pharmaceutics-13-00414-f002:**
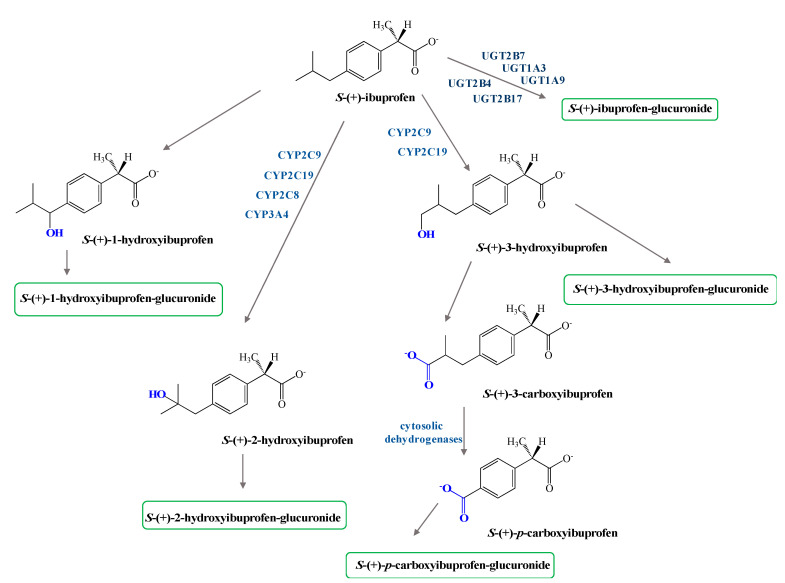
Oxidative metabolism of *S*-(+)-ibuprofen (UGT–uridine 5′-diphospho-glucuronosyltransferase; CYP–isoforms of cytochrome P450).

**Figure 3 pharmaceutics-13-00414-f003:**
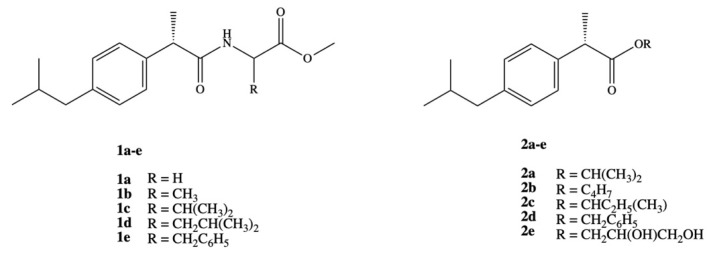
Amide and ester prodrugs of dexibuprofen.

**Figure 4 pharmaceutics-13-00414-f004:**
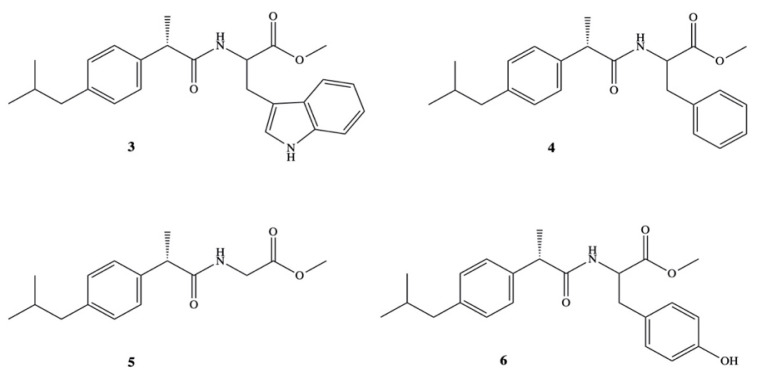
Amide prodrugs of dexibuprofen with l-tryptophan, l-phenylalanine, glycine, and l-tyrosine.

**Figure 5 pharmaceutics-13-00414-f005:**
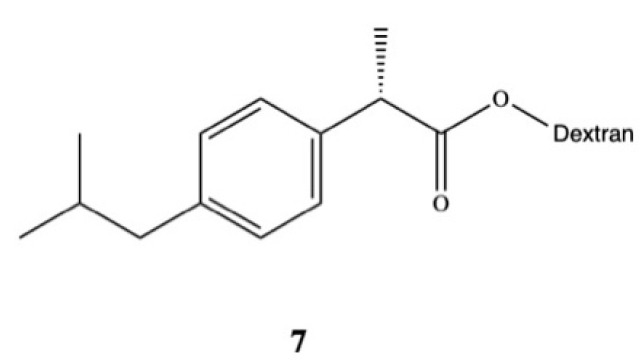
Dexibuprofen-dextran prodrug.

**Figure 6 pharmaceutics-13-00414-f006:**
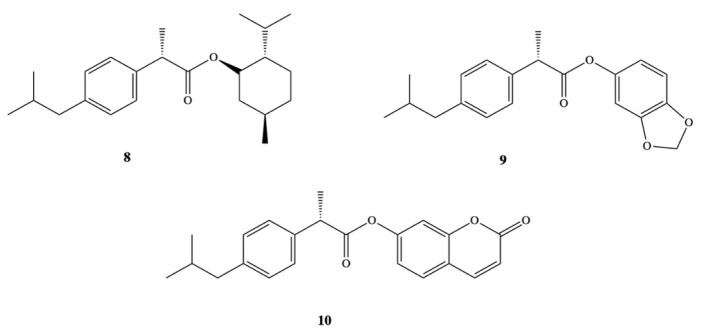
Dexibuprofen-antioxidant mutual prodrugs.

**Figure 7 pharmaceutics-13-00414-f007:**
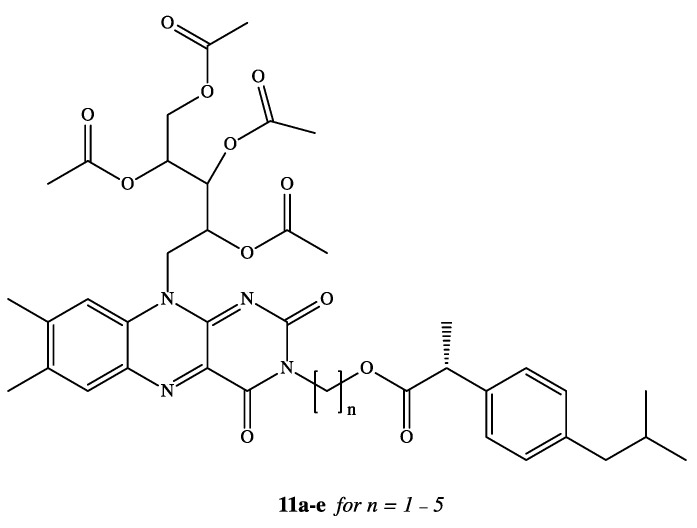
Dexibuprofen tetraacetylriboflavin conjugates.

**Figure 8 pharmaceutics-13-00414-f008:**
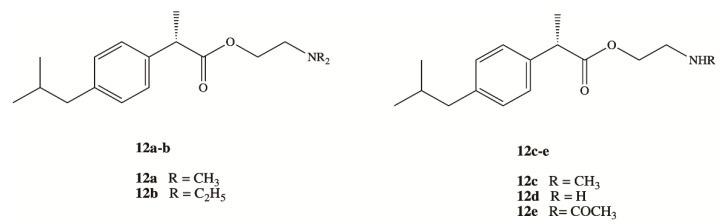
Dexibuprofen derivatives modified by ethanolamine related structures.

**Figure 9 pharmaceutics-13-00414-f009:**
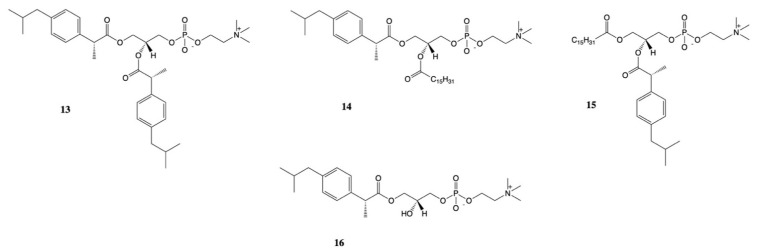
Conjugates of Dexibuprofen with phospholipids.

**Table 1 pharmaceutics-13-00414-t001:** Skin applied dexibuprofen (DXI) drug delivery systems.

Pharmaceutical Form	Physicochemical Characteristics	In Vitro Outcomes	In Vivo Outcomes	Ref
Transdermal patches Compounds: DXI, Polymeric excipients (ethyl cellulose and polyvinylpyrrolidone, plasticizer (di-*N*-butyl phthalate), Permeation enhancer (almond oil)	Uniform thickness (0.44 ± 0.02 cm) Low moisture uptake (7.87 ± 1.11 *w/w* %), Highly drug loading (100.0 ± 0.026%)	Ex vivo skin permeation studieS show 42% of DXI released in 4 h and 91% within 24 h	New Zealand rabbit model PatcheS were non-irritant Better pharmacokinetics: longer t_max_ (8 h) compared with DXI oral tablets (2 h) and increased half-life (10.51 h) against DXI oral tablets (3.50 h)	[69]
DXI microemulsion based transdermal reservoir patchesMicroemulsion compounds: ethyl oleate, Tween 80: PG	Microemulsion properties: size 119–221 nm Polydispersity index 0.35-0.56DXI loading >97% Stable for 6 months at 4 °C	Zero-order release rate Q_24h_ 79.13%; flow of 331.17 µg/cm^2^h	Model used: abino wistar rats No skin irritation Increased antiinflammatory effectivity against commercial hydrogel and ibuprofen emulsion gel	[70]
DXI Aloe vera trans emulgel	High DXI loading (78%) pH 7.56 Viscosity 112 CPs Flux 0.624 μg cm^−2^ h^−1^ Stable for the first 45 days	78% of the drug is released within 150 min	No skin irritation Superior anti-inflammatory activity (60.94%) than diclofenac gel (49.6%)	[71]
DXI emulgel. Compounds: gelling agent (Carbapol 940), penetration enhancers (Clove oil and Mentha oil), gel base	Stable for 3 months	In vivo release show 55.91–57.21% DXI released within 150 min Ex vivo permeation show DXI release 59.45–61.68% within 150 min	Comparable analgesic and anti-inflammatory activity against diclofenac gel	[72]
No-alcoholic transdermal DXI hydrogel. Compounds: pH modifying agent, antioxidants, water miscible solvent, HPMC, others	Stable for three months	Data not shown	Data not shown	[73]

**Table 2 pharmaceutics-13-00414-t002:** Oral DXI drug delivery systems (AUC, area under the curve; MRT, mean residence time; C_max_, maximum plasmatic concentration, T_max_, time to achieve maximum plasmatic concentration).

Pharmaceutical Form	Physicochemical Characteristics	In Vitro Outcomes	In Vivo Outcomes	Ref
Montmorillonite acid DXI composites	DXI loading of 298 mg/g	In vitro DXI released (92%) within 12 h in simulated intestinal fluid	Rat animal model. Better pharmacokinetic profile (AUC_0–24_ 644.49 μg/h/mL and MRT_0–24_ 7.65 ± 0.48 h) than DXI suspension (AUC_0–24_ 439.88 μg/h/mL and MRT_0–24_ 3.10 h). Increased bioavailability (154.11%) against commercial DXI	[73]
DXI chewable tablets	Preparation suing wet-co grinding of DXI adding mannitol and/or meglumine	DXI dissolution enhanced Mannitol based tablets showed prompt drug release Meglumine based tablets required crushing for fast drug release	Data not shown	[74]
DXI loaded β-cyclodextrin hydrogel nanoparticles	Nanoparticles size: 287 nm	DXI release higher than DXI tablets at pH 1.2 and 6.8	Animal model: Wistar albino rats Acute toxicity studies: no modification of behavioral, physiological, biochemical or histopathologic parameters were observed	[75]
DXI loaded hydroxypropyl- β- cyclodextrin (HPβCD) hybrid nanogels	Solubility enhancement of DXI confirmed Particle size 310.65 ± 18.75 nm Polydispersity index: 0.21 Zeta potential:−36.49 ± 2.34 mV	Highly porous and amorphous nanogels DXI release higher than DXI tablets at pH 1.2 and 6.8	Animal model: Wistar albino rats Toxicity studies: no modification of behavioral, physiological, biochemical or histopathologic parameters were observed	[76]
pH controlled DXI release hydrogel containing Dexibuprofen	Maximal gel swelling and drug release at pH 1.2.	Swelling and drug release pH-dependent Fast release at pH 1.2	Data not shown	[77]
DXI supermicro-pellet based dry suspensions	Pellets preparation using spray dry fluid bed coating technique Suitable stability (sedimentation rate 0.8 Hu/H) Aqueate flowability (θ 27°).	DXI release around 8 h being pH dependent	Data not shown	[78]
DXI loaded PLGA PEG nanoparticles	Nanoparticles size: 195.4 nm Polydispersity index: <0.1 Negative surface charge Stable for 2 months at 25 and 4 °C	100% nanoparticles uptaken by cells within 5 min Nanoparticles were able to cross trough and in vitro BBB model	Model: C57bl6 mice and APPswe/PS1dE9 transgenic Nanoparticles were effective for Alzheimer’s disease: inflammation and β-amyloid plaque reduction; behavioural improvement	[79]
DXI loaded chitosan nanoparticles	Particle size: 437.6 nm High entrapment efficiency (88.54%)	In vitro DXI release of 99.81% within 24 h	Data not shown	[80]
DXI Eudragit based microparticles	High entrapment efficiency (>70%)	In vitro DXI release at pH 1.2 < 21% while at pH 6.8 was high (around 60% within 8 h): gastro-resistant formulation developement	Data not shown	[81]
DXI Eudragit solid dispersed nanoparticles	Size: <300 nm	Improved dissolution rate	Animal model: sprague-dawley rats Improved pharmacokinetic parameters: AUC_0–24_ and C_max_ increased 4.6 and 5.7 times respectively	[82]
DXI loaded polymeric micelle based tablets	Size: 28.11 nm Polydispersity index: 0.15 Zeta potential −2.88 mV	Faster DXI release from the polymeric micelle based tablets (80.1% of DXI was released within 30 min) than the commercial tablet (35.35% within 30 min)	Human studies developed Pharmacokinetic studies: DXI polymeric micelles based tablets show better pharmacokinetic profile (AUC_0–24_ 407.45 µg mL h^−1^; C_max_ 20.99 µg/mL; T_max_ 1 h; MRT 12.79 h) than commercial tablets (AUC_0–24_ 71.91 µg mL h^−1^; C_max_ 12.94 µg/mL; T_max_ 2.75 h; MRT 10.53 h) Relative bioavailability was 160.15%	[83]

## Data Availability

Data is contained within the article.

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
