# Peer review of "Dexibuprofen Therapeutic Advances: Prodrugs and Nanotechnological Formulations"

_pharmaceutics, 2021, doi:10.3390/pharmaceutics13030414_

Round 1
Reviewer 1 Report
This article summarizes a novel formulations of DXI aimed to increase its therapeutic benefits minimizing the adverse effects, including its pharmaco logical information, pharmacokinetics, safety and therapeutic outcomes. Moreover, modified DXI drug delivery approaches are extensively discussed.
Major concern:
Considering space limitation, I suggest this article focuses on novel strategies for Dexibuprofen drug delivery and prodrug. The pharmacological, safety and therapeutic properties of Dexibuprofen should be introduced briefly.
Minor concerns:
- Figure 2: delete left down sign of body.
- "Marketed and clinical trials of Dexibuprofen medicaments" has little information and suggest to be deleted.
- References are too old.
Author Response
Dear Reviewer,
We would like to thank you for your comments. All your concerns had been addressed in the manuscript and modifications can be found in red colour. The report and corresponding responses to your suggestions can be found below.
According to the Reviewer 1 comments:
Considering space limitation, I suggest this article focuses on novel strategies for Dexibuprofen drug delivery and prodrug. The pharmacological, safety and therapeutic properties of Dexibuprofen should be introduced briefly.
Response: After careful revision of the manuscript, we agree that the sections entitled pharmacokinetics and safety and therapeutics of Dexibuprofen were too long. Therefore, we have merged them and reduced their content in order to provide only the essential information.
Minor concerns:
- Figure 2: delete left down sign of body.
Response: We corrected the Figure 2.
- "Marketed and clinical trials of Dexibuprofen medicaments" has little information and suggest to be deleted.
Response: We agree with the Reviewer and since this section was really short it has been merged with the previous one providing only essential information.
- References are too old.
Response: We have added some new references in order to provide updates.
We would like to express once again our thanks for very valuable comments, which help us to improve our manuscript. We hope that our explanations and corrections are sufficient, and will be accepted.
With kind regards,
Anna Gliszczyńska
Reviewer 2 Report
Overall, the review paper is well written. But, some of paragraphs are lengthy or are not necessary for the review purpose-prodrugs and nanotechnologies for dexibuprofen. For instance, sections 2~4 are already reviewed repetitively in the numerous review papers. Especially, section 4 is not relevant with the review topic unless they are clinical studies on dexibuprofen prodrugs/new nano-formulations. It should be better to mainly focus on the review topic. Moreover, the text should be properly separated by using comma (,). Edit the manuscript for readers.
Author Response
Dear Reviewer,
We would like to thank you for your comments. All your concerns had been addressed in the manuscript and modifications can be found in red colour. The report and corresponding responses to your suggestions can be found below.
According to the Reviewer 2 comments:
Overall, the review paper is well written. But, some of paragraphs are lengthy or are not necessary for the review purpose-prodrugs and nanotechnologies for dexibuprofen. For instance, sections 2~4 are already reviewed repetitively in the numerous review papers. Especially, section 4 is not relevant with the review topic unless they are clinical studies on dexibuprofen prodrugs/new nano-formulations. It should be better to mainly focus on the review topic. Moreover, the text should be properly separated by using comma (,). Edit the manuscript for readers.
Response: We completely agree with the reviewer comments and for this reason we have merged sections 2 to 4 and reduced it’s content in order to provide only essential information. In addition, more commas have been added in order to facilitate the understanding of the manuscript.
We would like to express once again our thanks to Reviewers for very valuable comments, which help us to improve our manuscript. We hope that our explanations and corrections are sufficient, and will be accepted.
With kind regards,
Anna Gliszczyńska
Round 2
Reviewer 1 Report
I feel that manuscript has been significantly improved and now warrants publication in Pharmaceutics.